# Universal Approach to Integrating Reduced Graphene Oxide into Polymer Electronics

**DOI:** 10.3390/polym15244622

**Published:** 2023-12-05

**Authors:** Elena Abyzova, Ilya Petrov, Ilya Bril’, Dmitry Cheshev, Alexey Ivanov, Maxim Khomenko, Andrey Averkiev, Maxim Fatkullin, Dmitry Kogolev, Evgeniy Bolbasov, Aleksandar Matkovic, Jin-Ju Chen, Raul D. Rodriguez, Evgeniya Sheremet

**Affiliations:** 1Research School of Chemistry & Applied Biomedical Sciences, Tomsk Polytechnic University, Lenina Ave, 30, 634050 Tomsk, Russiaellaijiah@gmail.com (I.B.); kogolev@tpu.ru (D.K.);; 2ILIT RAS−Branch of the FSRC “Crystallography and Photonics” RAS, 140700 Shatura, Russia; 3Department Physics, Mechanics and Electrical Engineering, Montanuniversität Leoben, Franz Josef Strasse 18, 8700 Leoben, Austria; 4School of Materials and Energy, University of Electronic Science and Technology of China, Chengdu 610054, China; jinjuchen@uestc.edu.cn

**Keywords:** reduced graphene oxide, thermoplastic polymers, graphene polymer composites, laser-induced polymer composites, flexible electronics

## Abstract

Flexible electronics have sparked significant interest in the development of electrically conductive polymer-based composite materials. While efforts are being made to fabricate these composites through laser integration techniques, a versatile methodology applicable to a broad range of thermoplastic polymers remains elusive. Moreover, the underlying mechanisms driving the formation of such composites are not thoroughly understood. Addressing this knowledge gap, our research focuses on the core processes determining the integration of reduced graphene oxide (rGO) with polymers to engineer coatings that are not only flexible and robust but also exhibit electrical conductivity. Notably, we have identified a particular range of laser power densities (between 0.8 and 1.83 kW/cm^2^), which enables obtaining graphene polymer composite coatings for a large set of thermoplastic polymers. These laser parameters are primarily defined by the thermal properties of the polymers as confirmed by thermal analysis as well as numerical simulations. Scanning electron microscopy with elemental analysis and X-ray photoelectron spectroscopy showed that conductivity can be achieved by two mechanisms—rGO integration and polymer carbonization. Additionally, high-speed videos allowed us to capture the graphene oxide (GO) modification and melt pool formation during laser processing. The cross-sectional analysis of the laser-processed samples showed that the convective flows are present in the polymer substrate explaining the observed behavior. Moreover, the practical application of our research is exemplified through the successful assembly of a conductive wristband for wearable devices. Our study not only fills a critical knowledge gap but also offers a tangible illustration of the potential impact of laser-induced rGO-polymer integration in materials science and engineering applications.

## 1. Introduction

A critical shift is underway in the pursuit of advancing flexible electronics. As we move beyond the realm of rigid circuitry, the quest for materials that can accommodate the demands of flexibility, strength, heat resistance, and electrical conductivity intensifies [1]. At the forefront of this transition are laser-assisted techniques, which offer precision, scalability, and the potential to revolutionize the fabrication of flexible electronic components based on polymer composites [2]. Laser-based approaches encompass different scenarios. Firstly, lasers can directly carbonize polymer surfaces, like polyimide (PI), polyethylene terephthalate (PET), polycarbonate (PC), polypropylene (PP), and polystyrene (PS), yielding laser-induced graphene (LIG) [3,4,5]. Alternatively, additives or coatings like carbon black, graphene, or metal nanoparticles can enhance polymer photosensitivity, enabling the creation of carbonized layers within the polymers [6,7,8,9,10,11]. Laser-assisted printing, such as laser-induced forward transfer, works regardless of material chemistry, offering improved adhesion, versatility, and an improved spatial resolution compared to traditional solvent-based coating methods such as inkjet or screen printing [2]. A less-explored avenue involves laser sintering, exemplified by Ko et al.’s work, where lasers were used to sinter metallic nanoparticles, deposited on the heat-resistant polyimide film by inkjet printing to create flexible organic field-effect transistors [12].

The adhesion challenge that impacts the durability of flexible electronics in harsh conditions, however, persists. Laser-induced polymer melting has emerged as a solution, enhancing interfacial adhesion. Notably, an approach denoted by authors as laser welding has been employed to create robust interfaces, such as carbon nanotubes/polycarbonate [13], and similar to interfacial adhesion of laser-reduced graphene oxide/PET for bioelectrodes [14]. This concept aligns with laser-assisted metal-polymer joining, where molten polymer fills defects in the metal to enhance adhesion [15,16]. This approach has led to the creation of robust conductive structures based on graphene and metal nanoparticles on PET surfaces [17,18], marking the initial steps in integrating nanomaterials into polymers using laser irradiation for flexible electronics.

Interestingly, a similar process has been demonstrated in polymer marking. Durable PP marking is achieved through near-infrared (NIR, 808 nm) laser curing of a polymer with a carbon particle ink layer [19]. The curing process involves ink migration within the polymer due to convection, resulting in a permanent image. To describe this process, the principles of laser melting models developed for coated systems can be applied [20], but complex polymer structures introduce limitations, necessitating consideration of factors like phase transition temperatures, chemical modifications, and heat transfer. Presently, the research into the mechanisms governing laser integration of nanomaterials, such as reduced graphene oxide (rGO), into polymers is to the best of our knowledge limited to just a few works cited here.

Our study explores the laser irradiation of a graphene oxide (GO) film on polymers, facilitating its reduction to conductive rGO and its integration into the molten substrate. While this approach has demonstrated its success with PET and other form of functionalized graphene in previous works [18,21], we sought to determine its applicability across various thermoplastic polymers. By fine-tuning laser processing parameters, we identified specific conditions that enabled the universal integration of rGO into all polymers studied, regardless of their composition or crystallinity. This observation aligns with prior research on laser patterning of thin polymer films on silicon, demonstrating profiles somewhat independent of polymer chemistry [22]. We not only elucidate the fundamental mechanism behind this universal laser-driven rGO integration but also apply it to polymers relevant to additive manufacturing. These polymers include those used for 3D printing, enabling cost-effective, robust, and versatile production of flexible electronics. To illustrate the practical implications of our work, we present the creation of a conductive wristband for a wearable smartwatch, a product of the rGO laser-induced integration, and subsequent thermoforming. Our graphene-polymer composite endures the thermoforming manufacturing process, offering a path for further optimization and practical application of the process and opening new ways for flexible and robust polymer electronics.

## 2. Materials and Methods

### 2.1. Polymers Preparation

The following polymers were selected for this work: polyethylene terephthalate (PET), polyethylene terephthalate glycol (PETG), acrylonitrile butadiene styrene (ABS), nylon, poly-L-lactic acid (PLLA), thermoplastic polyurethane (TPU), styrene-butadiene copolymer (SBS) and polyvinylidene fluoride (PVDF). PET was a commercially available sheet with a thickness of 0.6 mm. The TPU, PVDF, and PLLA sheets were prepared by the method of Fused Deposition Modeling (FM) 3D printing. The remaining polymers, PETG, ABS, SBS, and nylon, were prepared by the thermal molding method at 240 °C at a heating plate for 15 min from 3D printing filaments (Bestfilament, Tomsk, Russia).

### 2.2. GO Deposition

Graphene oxide 1 mg/mL dispersion was prepared from 4 mg/mL dispersion from Graphenea, Gipuzkoa, Spain. Before deposition, the dispersion was sonicated for 10 min. The deposition of the GO film onto polymer sheets was carried out using a drop-casting method. To minimize the impact of “Coffee Rings” and ensure uniform processing of the GO on the polymer substrate, we focused on central areas of the substrate, avoiding edge regions where this phenomenon is prevalent. This method ensured more consistent GO areal density, as evidenced in Appendix A, thus minimizing dispersion irregularities due to edge agglomeration. The area density of the GO was 180 µL/cm^2^. A photograph of the GO/PET and dry GO/PET can be found in Appendix A. The contact angles between the GO dispersion and various polymers varied, yet the use of this specific amount of GO ensured the formation of complete coverage films on all surfaces.

### 2.3. Laser Processing

A diode laser, emitting at a central wavelength of 438 nm, operated quasi-continuously was used for GO reduction. The laser beam was focused on the surface with a 10 × 0.28 NA objective. The laser spot size on the sample surface was determined to be ~30 × 100 µm^2^ with a rectangular shape. The laser spot side was turned at an angle of about 45° with respect to the scanning direction. Laser power was adjusted to 24, 40, 55, and 78 mW (0.8 kW/cm^2^, 1.33 kW/cm^2^, 1.83 kW/cm^2^, and 2.6 kW/cm^2^ respectively) after the objective through a series of neutral optical filters. The distance between lines was 50 μm. The selection of XY stage movement speed during the reduction process was based on obtaining rGO/polymer composite with the lowest resistance. Speeds of 25 mm/min and 50 mm/min were tested, both yielding positive outcomes. However, 25 mm/min was selected for further study, as it offered the advantage of capturing finer details during high-speed video recording.

### 2.4. Ultrasonication of Prepared Samples

To test the rGO adhesion to the polymer, the rGO/polymer samples were sonicated in a 120 W ultrasound bath filled with distilled water for 1 min.

### 2.5. Heating Model of rGO/Polymer Structures

A simulation of laser processing of the GO on polymer surfaces was performed using COMSOL Multiphysics software version 6.1, allowing for a comprehensive comparison between experimental and theoretical data. All the relevant material properties used are provided in Appendix A.

### 2.6. High-Speed Camera Recording

Laser processing videos were recorded with a Phantom Miro C110 high-speed camera. The video recording settings were configured with a minimum frame rate of 200 frames per second, providing the necessary quality to assess the behavior of GO, rGO, and polymer.

### 2.7. Electrical Measurements

Resistance measurements were carried out in a 4-probe configuration using Potentiostat/Galvanostat P-45X with FRA-24M impedance modulus (Electrochemical Instruments, Chernogolovka, Russia) in galvanostatic mode and MST 4000A microprobe station (MS Tech, Seoul, Republic of Korea). Probes were set at a square with a 300 μm side. Measurements were conducted before and after sample sonication. Due to sample inhomogeneity after sonication, the geometry of the sample is not considered suitable for calculating the sheet resistance correctly, and further, the samples are only distinguished as conductive or non-conductive.

### 2.8. Optics

Optical images of samples’ surfaces were captured using a digital camera coupled with a microscope and 5× objective.

### 2.9. SEM and EDX

The SEM and EDX images were made with a Mira 3 (Tescan, Brno, Czech Republic) scanning electron microscope and 500× magnification.

### 2.10. Fe_3_O_4_/rGO Experiment

The Fe_3_O_4_/rGO/polymer samples were fabricated using the same protocol as for rGO/polymer samples. The concentration of Fe_3_O_4_ nanoparticles (Advanced Powder Technologies, Tomsk, Russia) added to the GO dispersion was 0.1 mg/mL.

### 2.11. XPS

A Thermo Fisher Scientific (Waltham, MA, USA) XPS NEXSA spectrometer with a monochromated Al K Alpha X-ray source working at 1486.6 eV was used for XPS. For the high-resolution spectra, pass energy was 50 eV and energy resolution was equal to 0.1 eV. The spot area was 400 µm^2^. The flood gun was used for the charge compensation.

### 2.12. DSC/TGA

The TGA-DTA analysis was carried out with a Q600 Simultaneous TGA-DTA analyzer (TA Instruments, New Castle, DE, USA) in an open alumina crucible under a dynamic air atmosphere with a 100 mL/min flow rate, from ambient temperature to 800 °C at a heating rate of 10 °C/min.

The differential scanning calorimetry (DSC) experiments were carried out using the DSC Q2000 (TA Instruments, USA) with aluminum Tzero pans with leads at a heating rate of 10 °C/min and with a nitrogen flow rate of 20 mL/min. Temperature calibration was carried out using melting points of pure indium, tin, and lead. Heat flow was calibrated with a sapphire standard.

## 3. Results and Discussion

The work concept is illustrated in Figure 1. Initial tests indicated partial success in laser-induced reduced graphene oxide integration (Figure 1a), with effectiveness varying across polymer types, likely due to the nuanced interactions depicted in Figure 1b. However, our detailed investigation revealed specific laser parameters that consistently achieved the rGO integration across all examined polymers, despite their diverse chemical and structural properties (see Figure 1c). Here, we discuss the results from eight distinct polymers and four laser power settings, solving the underlying mechanisms that enable the universal integration of the rGO into thermoplastic polymers and elucidating the specific conditions that govern these integration regimes.

Our analysis begins with an evaluation of the thermal characteristics of the polymers, focusing particularly on destruction and phase transition temperatures, based on thermogravimetric analysis and differential scanning calorimetry. These critical thermal points were then compared with peak temperatures attained during laser exposure, as predicted by numerical simulations. This comparative approach allowed us to establish a correlation between simulated temperature profiles, inherent thermal properties of the polymers, and the structural and electrical attributes of the resulting GO/polymer films post-laser treatment.

The surface morphology of the laser-treated samples was investigated using scanning electron microscopy (SEM), providing detailed imagery of the rGO and polymer changes induced by the laser processing. The extent of the rGO distribution within the various polymers was quantitatively assessed through the detection of elemental markers via energy-dispersive X-ray spectroscopy (EDX). This comprehensive analysis not only elucidates the conditions facilitating universal rGO integration but also offers profound insights into the transformative potential of this methodology for materials engineering.

### 3.1. Thermal Analyses Reveal Polymer Phase Transitions

Our investigation started with the assumption that integration is driven by the photothermal transduction of laser energy in the GO/polymer system. Thus, we focused on exploring the thermal properties of a range of polymers widely used in real-world applications [23,24,25,26,27]. We used thermogravimetric analysis (TGA) and differential scanning calorimetry (DSC) (Appendix A) to determine the key temperatures where phase transitions occur: glass transition *T_g_*, melting point *T_m_*, and decomposition *T_d_* (Appendix A). These parameters define the polymer’s behavior upon heating. The comparison of *T_d_* lets us roughly estimate the upper threshold value of laser power, below which the polymer does not undergo decomposition. For example, while styrene-butadiene copolymer SBS (*T_d_* of 405 °C) can withstand 78 mW laser power almost without substantial damage, the same power could disintegrate PLLA (*T_d_* of 316 °C). On the other hand, for crystalline polymers, *T_m_* is considered to be the minimal temperature that should be reached under laser processing to ensure composite formation or in other words a lower threshold value.

To simplify our approach, we categorized polymers into three main groups: crystalline, semi-crystalline, and amorphous. High-crystalline polymers, characterized by distinct melting points, transition from an organized to a random molecular structure upon reaching *T_m_*. Contrarily, semi-crystalline and amorphous polymers do not exhibit *T_m_*. For classification, we either used the presence or absence of a melting point as a benchmark. In our DSC curves (Appendix A), pronounced peaks demonstrated high crystallinity (as seen in nylon, PVDF, PET, and PLLA, unclear peaks denoted semi-crystallinity (observed in TPU), and the absence of peaks indicated amorphous structure (like in ABS, PETG, and SBS).

The GO water dispersion was drop-casted on the polymer surface. We laser-treated the GO on these polymers at varied powers to discern any links between their thermal profiles and laser-induced structural changes. After the laser irradiation, we were able to estimate the degree of conductivity using electrical resistance measurements, both before and after a brief 1-min sonication in water. For accuracy, we employed a qualitative 4-point probe method, avoiding quantitative assessments due to inconsistent data arising from laser-induced irregularities, evident in Figure 2’s trench patterns (marked in Figure 2a). In this figure, tick marks and cross marks (further highlighted with red in Appendix A) respectively denoted samples retaining (Figure 2b,d,f,h) or losing conductivity (Figure 2g,l) post-sonication.

After sonication, regions with weak adhesion, visible as bubbles beneath the rGO layer (Figure 2i,k and Appendix A), were removed from the rGO/polymer composite, unveiling the underlying polymer substrate (Figure 2j,l). This delamination phenomenon was particularly noted at lower laser powers for all amorphous polymers, specifically, ABS, PETG at 24 and 40 mW, and SBS at 24 mW, as well as for the crystalline ones, nylon, and PET, at the same power levels. However, the PETG and SBS samples remain conductive because of reduced delamination despite clear signs of delamination in the optical images. To conduct a more detailed investigation, we measured the resistance of individual lines on each polymer, as shown in Appendix A. We discovered that lines created using 55 mW (for all polymers except PETG and PLLA) and 78 mW (for all polymers) exhibited conductivity, suggesting that higher laser powers also facilitate the formation of conductive composites. However, no conductivity was observed in the large-area rGO/polymer composites at 78 mW. Given a 50 μm step between lines, overlapping at the edges occurred, leading to double processing of certain composite sections. Notably, lines produced at 24 and 40 mW did not show conductivity, unlike their large-area counterparts. This indicates that at lower powers, repeated processing contributes to forming conductive composites, while at higher powers, it impedes conductivity, necessitating more precise optimization of line spacing.

Among the crystalline polymers used, PET and nylon have the highest melting points (242 and 177 °C respectively). This led us to assume that the laser power at the lower values was insufficient to heat the polymer to its melting threshold, resulting in rGO peeling off during sonication and a subsequent drop in electrical conductivity. Conversely, crystalline counterparts with lower melting temperatures, the PVDF and PLLA (151 and 150 °C, respectively), along with semi-crystalline TPU (with a *T_m_* of 180 °C), retained stable, conductive structures under all tested power conditions. This observation highlights the critical role the melting point assumes in the interfacial bonding strength between rGO and the (semi-)crystalline polymers.

The decomposition temperature is another critical factor in conductive composite formation, especially evident with PLLA’s lower *T_d_* of 316 °C. At 55 mW laser power, PLLA showed signs of decomposition, highlighted by unique morphology but no conductivity, emphasizing the need for a precise thermal balance during laser integration. As seen when increasing power to 78 mW, excessive power disrupts this balance so that all polymer coatings lack conductivity. One exception is the rGO/PVDF with the PVDF having the highest *T_d_*, reinforcing the idea that polymer destruction is the reason the conductivity is lost for high laser powers.

This underscores the fine line between using adequate laser power to induce conductivity and exceeding a polymer’s thermal threshold, leading to decomposition. The process demands precise laser calibration to respect each polymer’s thermal characteristics, ensuring conductivity without damaging the material’s integrity.

### 3.2. Numerical Simulation of Laser-Induced Heating

To confirm the assumption made in the previous section on the correlation between polymers’ thermal properties and the stability of the final structures, it is necessary to have an idea of the temperatures reached under laser processing. We designed a numerical model using the finite element method (FEM), to calculate the temperature distribution for the GO/polymer system reached under laser irradiation and its gradient along the three dimensions. We used literature values for the polymers’ properties, such as density, thermal conductivity, and specific heat capacity (Appendix A). The same parameters for the GO were also taken from literature data [28,29,30,31,32,33,34]. The maximum surface temperatures for 0.25 s irradiation time are shown in Figure 3a. The irradiation time was selected based on laser scanning speed and effectively considers the time the laser shines on one spot, not the heat generated while scanning neighboring areas. This means that the resulting temperature calculated is likely underestimated; nevertheless, it gives an idea of the lower limit of the temperature that can be reached.

First, we focus on crystalline polymers (nylon, PVDF, PLLA, and PET). The surface temperature of the GO/PLLA composite at 55 mW processing, reaching 368 °C, surpasses PLLA’s *T_d_* (316 °C). This confirms the earlier hypothesis, that rGO/PLLA composite is not conductive due to thermal decomposition of the polymer. The temperatures of GO/PLLA during laser processing with 24 and 40 mW greatly exceed PLLA’s melting temperature (*T_m_*), facilitating substrate melting and rGO adhesion. Similar findings apply to PVDF. However, in the case of PET, laser processing with 40 mW induces heating to the temperature only slightly above the melting point (5 °C above) leading to no composite formation, while for PVDF and PLLA induced temperatures are more than 100 °C higher than the melting point. These findings suggest that the heat generated during laser processing should be high enough to greatly exceed the polymer’s melting point. At the laser power of 78 mW, where calculations showed that the temperature for all the polymers is higher than *T_d_*, no electrically conductive structures were formed with one exception of PVDF. This allowed us to formulate a general conclusion: the formation of a conductive composite between rGO and high-crystalline polymers during laser processing is possible under the following conditions:(1)$$T_{m}\ll T_{L}<T_{d}$$
where *T_m_* represents the polymer’s melting temperature, *T_L_* denotes the maximum temperature reached under laser irradiation, and *T_d_* corresponds to the polymer’s decomposition temperature. Nylon is an exception to the expected pattern: it fails to form a stable structure at 40 mW, even though it satisfies condition (1). This anomaly is likely linked to nylon’s significantly higher heat of fusion (Δ*H_f_*), calculated at 58.9 J/g, in contrast to other crystalline polymers in our study, such as PLLA (7.9 J/g) and PVDF (14.1 J/g). The elevated Δ*H_f_* of nylon implies a greater heat requirement for its phase transition, resulting in a comparatively lower temperature at the material’s surface during laser exposure. This observation illustrates the intricate dynamics of the laser-GO-polymer interactions and the importance of incorporating additional parameters into their simulation models for a more accurate prediction of their behavior.

Amorphous polymers present a unique challenge due to the absence of a defined melting point. Instead, their behavior is guided by the glass transition temperature (*T_g_*), at which they transition to a rubbery state, gaining flexibility and exhibiting characteristics like reduced viscosity and enhanced flow [35]. During laser processing, all samples are heated above *T_g_*, placing them in this rubbery state. The viscosity is also defined by the fragility index which shows how fast the polymer becomes liquid with increasing temperature (see Appendix A).

Intriguingly, increasing the laser power seems to promote the laser-induced integration of amorphous polymers. This can be attributed to heightened molecular mobility at elevated temperatures, leading to decreased viscosity [36] and more effective merging of the rGO with polymers, caused by Marangoni flows prevalent at high-temperature gradients.

Our study suggests that the amorphous polymers with low fragility (ABS) do not form stable conductive structures at lower powers because of lower molecular mobility near the glass transition point, while high fragility polymers (TPU, SBS, and PETG) form stable structures due to lower viscosity with temperature increase (see Appendix A). This hypothesis, while compelling, requires additional studies focusing on polymer rheology for confirmation. Moreover, akin to crystalline and semi-crystalline counterparts, amorphous polymers do not become electrically conductive at high laser powers (78 mW) due to the thermal decomposition when surpassing their decomposition temperature, setting a practical upper boundary for laser power application.

We summarize all the findings of this section in Figure 4. At low laser power, we observe the electrical conductivity of samples that confirms the laser reduction of the GO (Figure 4a). However, the adhesion is insufficient to provide mechanical robustness. Conversely, high laser power compromises material integrity. Therefore, the optimal conditions hinge on the polymers’ properties: crystallinity, melting point, glass transition temperature, and decomposition temperature (Figure 4b,c).

### 3.3. Optical Microscopy Analysis

The simulation results depicted in Figure 3b and Appendix A reveal the presence of radial temperature gradients induced during the laser processing of GO/polymer. To gain insights into the transformations of GO/polymer in real-time and to elucidate the materials’ response to laser irradiation we integrated a high-speed camera into the laser processing setup and recorded the process at 200 fps. This system allowed for a more detailed analysis of morphological changes with sufficient spatial and temporal resolutions.

We selected representative keyframes from high-speed video recordings, showing different types of surface modification for all laser powers, see Figure 5a and Appendix A. For lower laser powers of 24 and 40 mW, we noticed two distinct regions: GO expansion at the laser spot due to gas release and the lifting of the GO film around this spot. The GO film expansion results from several processes: gas release from the melted polymer and GO reduction itself (oxygen-containing groups being removed as gas species), absorbed water evaporation, polymer thermal expansion, and increasing interlayer separation in rGO [37,38,39].

After removing the residual GO by ultrasonication, we observed dark laser-irradiated lines that remain intact and therefore are attributed to the rGO/polymer composite formation. Around them, there is a well-defined polymer modification area that we attribute to a melt pool. The melt pool width increases with laser power as indicated in Figure 5b for the case of PET (all other polymers are presented in Appendix A). Qualitatively, these observations are consistent with the simulation and experimental results in Figure 3b and Figure 5b.

At higher powers (55 and 78 mW) a trench is formed in the middle of the rGO region. We attribute the appearance of this trench to more extensive convective flows, driven by higher temperature gradients than for 24 and 40 mW laser powers. This convection coupled with recoil pressure, pushes the material away from the center [37]. However, we should also consider the possibility of the rGO ablation from the line center where the temperature is the highest. This leads us to divide the polymers into two sets: one showing this trenching effect (ABS, PET, PLLA, PVDF, and SBS) and another (nylon, PETG, and TPU), displaying surface melting at 55 and 78 mW. In the latter case, high-speed recording shows that the polymer flows into the previously formed trench, similar to the Marangoni flow observed in selective laser sintering of Ag nanoparticles [40]. These contrasting responses highlight the distinct rheological and thermal qualities of each polymer, which dictate their behaviors during laser exposure.

The data compiled from various optical analysis methods are summarized in Appendix A and the histogram in Appendix A. This histogram allows us to compare the melt pool width, the GO lift area, and the composite structure width from both top-view and cross-sectional perspectives (Appendix A). These cross-sectional analyses reveal polymer redistribution from the laser line to the peripheral edges, due to Marangoni flow. This observation serves as a clear indication of polymer melting, which is most prominent in the case of the PLLA. Interestingly, there is a notable similarity in the measurements of the GO lift width obtained from the video recordings and the melt pool width derived from the top-view images. This parallel suggests that the lifting of GO is likely a consequence of processes occurring within the polymer. Potential contributors to this phenomenon include water evaporation, gas release, or the GO’s inability to integrate due to the surface tension coefficient of the molten polymer. This surface tension is presumably higher at the peripheries of the melt pool, hindering the integration of GO into the polymer bulk, while it is reduced near the heat source (laser), facilitating GO intermixing. These considerations lead us to hypothesize that the polymer flow dynamics, particularly around the laser-irradiated zone, actively contribute to the observed lifting of the GO.

Moreover, our observations post-GO deposition indicate that the flakes form a stable film, likely bonded by hydrogen bonds in functionalized graphene and possibly π-π interactions. However, after laser treatment, we noted inhomogeneous rGO surfaces on the substrates, as evident in Appendix A. This inhomogeneity is attributed to laser beam scanning and Marangoni flow effects, leading to the formation of trenches with higher rGO concentrations along the borders. The residual rGO, visible as dark flakes post-washing (see Figure 5b), further supports this. The system’s electrical conductivity indicates multiple conductive pathways, suggesting the agglomeration or sintering of the rGO flakes. This agglomeration persists even after the removal of oxygen-containing groups, likely due to strong van der Waals forces [41].

### 3.4. Elucidating Mechanisms behind the Formation of Electrically Conductive Composite via SEM/EDX and XPS

Different laser powers (from 24 to 55 mW) resulted in the formation of conductive and stable coatings for multiple polymers. However, the conductive layer may be formed through different pathways as laser power increases (see summary in Figure 6a). One could imagine two scenarios for the formation of the conductive composite layer: (1) polymer integration as discussed above, and (2) the carbonization of the substrate that may play a role in electrical conductivity as it happens in the case of LIG [42]. To address this question, we turned to the SEM/EDX analysis to elucidate these two distinct mechanisms.

However, the challenge arises from the fact that the GO and the substrates are carbon-based, making it difficult to use elemental tracking methods like EDX to investigate the integration process. To overcome this limitation and distinguish between the conductive layer formation mechanisms, we conducted laser processing of a GO film mixed with Fe_3_O_4_ nanoparticles (FeNPs) at a concentration of 0.1 mg/mL. By mixing FeNPs with the GO film, we could track the fate of these particles during the laser processing, which, in turn, reveals details about rGO’s behavior and its integration with the polymer matrix. One must keep in mind, however, that adding the FeNPs may shift power thresholds and this particular effect was not investigated separately, but the FeNPs content was chosen to be 6.4% in the initial mixture. This amount is high enough to be detected in EDX even though no effect on integration was observed based on visual inspection. Therefore, we assume that the key mechanisms remain mainly the same with no significant influence from FeNPs. Subsequently, the sample was sonicated for 1 min after laser processing, followed by EDX analysis to confirm whether the FeNPs remained attached to the surface.

After laser processing and sonication, the samples were examined using SEM/EDX to assess the presence of iron from the FeNPs on their surfaces. The SEM images for the rGO/Nylon and rGO/PVDF are shown in Figure 6b–d,h–j, respectively. The corresponding EDX maps, depicting iron distribution, are displayed in Figure 6e–g,k–m. A detailed summary of the iron content analysis is presented in Appendix A. The surface iron content ranged between 0.7% and 5.2%, indicating some Fe loss, which reflects the extent of the rGO integration into the polymer surfaces as a portion of loosely bound material was removed during sonication. These results suggest partial integration of iron oxide NPs into the polymer matrix during laser processing at powers of 24 to 55 mW for all tested polymers, except PLLA at 55 mW where surface degradation precluded the experiment. This observation supports the hypothesis that better composite formation correlates with higher polymer temperature, as evidenced by the increased Fe content at higher laser powers in nylon (Figure 6b–g). However, several polymers exhibited Fe loss at higher powers, indicating possible partial ablation (Figure 6h–m), thus supporting the integration mechanism (#1) while not ruling out carbonization (#2) (Figure 6a).

The XPS analysis was performed to explore the possibility of carbonization. The formation of a conductive network implies a chemical restructuring, reflected by the creation of C==C (sp^2^) bonds. Given the complexity of separating C-C and C=C peaks, we analyzed the combined atomic content of these bonds in the C-region (280–294 eV). These results are summarized in Figure 6d. During the reduction of GO to rGO on glass, the CC (sp^2^+sp^3^) content increased by 18.6% at 55 mW laser power, with no further increase at higher powers as the C–O–C and O–C=O bond content remained stable. Considering the lower CC bond content in raw polymers compared to the rGO, a 44.5% threshold (observed in rGO at 55 mW, marked by a red line) can be considered the limit for GO reduction in the GO/polymer system. Any further increase in CC bond content suggests polymer carbonization, which depends on the processing temperature related to substrate properties. Notably, CC bond content did not increase at 78 mW for any polymers, correlating with material decomposition at *T_d_* as simulated in Figure 3a.

In practical applications, these insights are invaluable. Understanding that lower laser powers favor rGO integration can guide the operational parameters for the fabrication of conductive composites, ensuring the preservation of the polymer matrix’s integrity while optimizing conductivity. Conversely, recognizing that a higher laser power of 55 mW promotes substrate carbonization over the rGO integration helps in applications where the carbonized structure itself is the desired outcome, such as in the creation of LIG.

### 3.5. Laser-Induced Composite Formation Mechanism

Further, we summarize the key processes and their influence on the laser-induced rGO/polymer composite formation to formulate the mechanism behind this phenomenon (Figure 7). One of the key processes is light-matter interaction with dark GO and rGO films, mostly represented by absorption. The absorbed light heats the GO and part of this thermal energy dissipates to the polymer substrate underneath. For thermoplastic polymers, we consider that once a polymer is heated the hydrogen bonds, holding the polymer chains together, start dissociating. Hence, the polymer experiences glass transition at *T_g_* and/or melts at *T_m_* and degrades (typically via pyrolysis reaction resulting in carbonization) [43] at *T_d_*. Based on the high-speed video recordings discussed above, melt pool formation is a necessary condition for the integration of the rGO into the polymer. The movement of the polymer and rGO in the melt pool caused by convective flows can be described by the Marangoni effect [44]. The strength of the capillary flow is determined by the surface tension dependence on the temperature, temperature gradient, and viscosity (for polymers it is strongly affected by temperature) [45], thermal diffusivity, and characteristic width of the melt pool [45].

In turn, the GO laser reduction also exhibits several distinct stages. At 24 and 40 mW the reduction of the GO film is observed regardless of the polymer substrate used. In addition, we could clearly see that the surface topology changes not only due to gas release but also due to the movement of the polymer surface underneath the GO. At a higher power (40 mW) reduced area and, the width of the GO lifts around it became broader.

At 55 mW, the ablation threshold is reached, causing GO removal along the laser’s path. Different polymers showed varied responses where nylon, PETG, and TPU exhibited liquefied polymer spilling out from the ablation region along the path of the laser beam on top of the rGO film, while other polymers ABS, PET, PLLA, PVDF, and SBS do not display such effect (see high-speed recordings in the SI). The width of the modified region expands with increasing laser power for almost all polymers, except ABS, PETG, and SBS, due to heat dissipation. Going back to EDX data we can say that the carbonization of some polymers takes place. As depicted in Figure 6, EDX showed that on some polymers, such as PVDF and Nylon, there are only trace amounts of Fe atoms left in the structure after laser processing, even though the material is conductive. It leads us to conclude that for the laser power of 55 mW, the GO film containing FeNPs was ablated, so the material conductivity is explained by polymer carbonization. In contrast, at laser powers below 55 mW, the films exhibit a substantial presence of iron, providing indirect evidence of rGO integration.

Taken together, the evidence suggests that the primary condition for the successful creation of a laser-induced rGO/polymer composite is the thermoplastic characteristics of the polymer. However, it is important to note that thermoplastic polymers display diverse thermal behaviors, including varying glass transition temperatures, decomposition temperatures, and melting points. As such, condition (1) discussed above is crucial to guarantee the formation of the composite.

Here is a simplified breakdown of the steps and factors involved in laser-induced composite formation:GO Photothermal Heating: Laser irradiation leads to the absorption of light by the GO, initiating its thermal response. This is the primary interaction between light and matter in this context (Figure 7a).Transformation of the GO to rGO: The absorbed thermal energy prompts the reduction of the GO to rGO (reduced graphene oxide), enhancing its optical absorption properties and consequently, its heating. This transition is critical as the rGO possesses the electrical conductivity desired for the final composite (Figure 7a,b).Heat Transfer to Polymer: The heat generated in the rGO is transferred to the underlying polymer substrate, elevating its temperature, and affecting its physical state (Figure 7b).Polymer Melting: Upon reaching its melting temperature, the thermoplastic crystalline polymer transitions from solid to liquid. This phase change is essential for the subsequent integration of the rGO layers into the polymer matrix (Figure 7c). In the case of amorphous polymers, the processing temperature should exceed the glass transition point to provide the polymer in a rubbery state with increased molecular mobility.Intermixing of rGO and Molten Polymer: In the liquid state, convective flows (described by the Marangoni effect) facilitate the movement and mixing of the rGO into the molten polymer. This capillary flow is influenced by various factors including temperature-dependent surface tension, viscosity, thermal diffusivity, and the melt pool’s characteristics (Figure 7c).Composite Solidification: As the mixture cools, it solidifies, encapsulating the rGO within the polymer matrix. This solidification locks the conductive rGO particles in place, forming a continuous, electrically conductive pathway within the composite. Notably, the solidification of amorphous polymers is defined by glass transition temperature and molecular dynamics around this point (so-called fragility).Threshold of Polymer Degradation: If the laser power continues to increase such that the substrate’s temperature reaches its degradation temperature (*T_d_*), the polymer undergoes pyrolysis. This leads to carbonization, which can contribute to electrical conductivity but also compromises the structural integrity of the polymer matrix. At this stage, the conductive pathways are more likely due to the carbonized substrate rather than the integrated rGO.

### 3.6. Application: Thermoforming a Wearable Wristband

Our study’s insights into the integration of GO into polymers have significant practical implications, particularly in flexible electronics. We showed this with the creation of a durable, wearable wristband (the thermoforming process is shown in video 1 in the Appendix A). What sets this application apart is the utilization of thermoforming, a process that involves subjecting the composite material and polymer to heat to mold them into a desired 3D shape, as depicted in Figure 7d. This process shows the remarkable resilience of the material, emphasizing its potential for demanding applications.

Another crucial aspect of this application demonstration is our capacity for scaling up the manufacturing process. The extended surface area of the wristband dictated a need for increased production efficiency. Armed with our comprehensive understanding of the various underlying processes, we successfully accelerated the operation while fine-tuning the laser power. Specifically, we employed a 78-mW laser power at a 5 mm/s laser processing speed during the rGO integration phase. This deliberate laser power adjustment at high scanning speed ensured the establishment of electrical conductivity consistent with those attained at lower laser power configurations.

Through the practical application of our insights into the GO integration with polymers, we unlock vast possibilities for an array of flexible and wearable electronic devices that can enhance the functionality and comfort of these pervasive technologies.

## 4. Conclusions

This study’s innovation centered on the discovery and application of universal parameters for integrating reduced graphene oxide into various thermoplastic polymers using laser technology. The universal applicability of our findings is demonstrated by the successful adaptation of these parameters across diverse polymer types, transcending the limitations of polymer composition and crystallinity. This has significant implications for advancing material technologies in areas like wearable and flexible electronics, illustrating the broad potential of our research in material science. Notably, we found that laser power densities of 24 and 40 mW (0.8 and 1.33 kW/cm^2^, respectively) were effective in reducing graphene oxide without causing ablation. For polymers with melting points below 200 °C, an adhesion mechanism like in laser-assisted polymer joining was observed, as evidenced by energy-dispersive X-ray spectroscopy using iron nanoparticles to trace the rGO integration. In contrast, higher laser powers of 55 and 78 mW (1.83 and 2.6 kW/cm^2^) caused ablation of the GO at the laser spot, with subsequent reduction in the surrounding areas. This led to the creation of conductive surfaces at 55 mW (1.83 kW/cm^2^) for most polymers, with the notable exception of the PLLA, which suffered adverse effects. However, at 2.6 kW/cm^2^, most structures were non-conductive, except those on the PVDF polymer. Overall, our findings not only enhance our understanding of the formation mechanisms of laser-induced rGO/polymer composites but also demonstrate the significant potential in developing state-of-the-art wearable and flexible electronic devices. The capability to finely tune material properties through laser technology indicates new possibilities in diverse industrial and technological applications.

## Figures and Tables

**Figure 1 polymers-15-04622-f001:**
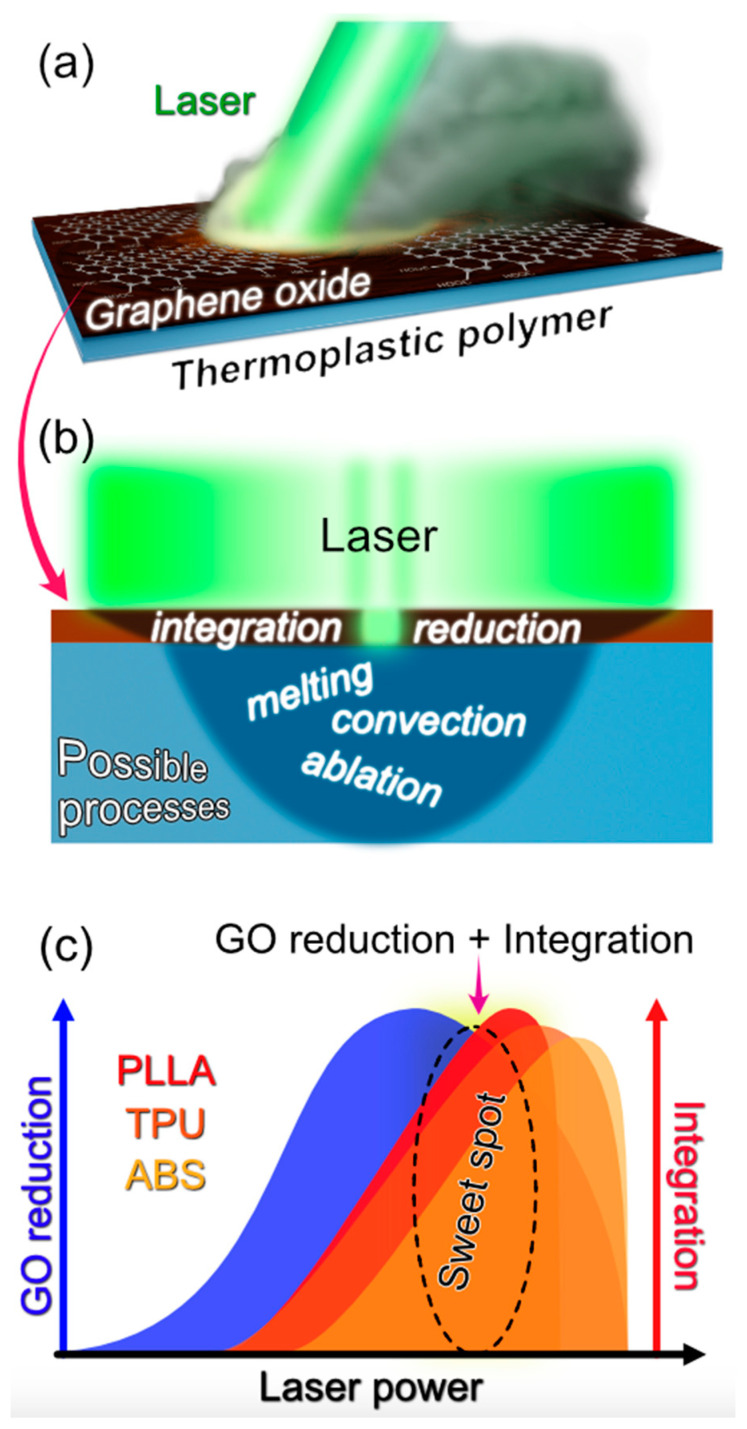
Schematic Overview of Laser-Induced Graphene-Polymer Composite Formation. (**a**) Depicts the initiation of a conductive composite via laser processing. (**b**) Categorizes the various phenomena occurring during the laser treatment of the GO with the polymer matrix. (**c**) Illustration of the effective laser power range for the GO reduction (blue region) and its concurrent integration (red and orange regions) with diverse polymer substrates. The orange-red region marked “Sweet Spot” represents the laser power threshold where the GO reduction and polymer integration occur most effectively.

**Figure 2 polymers-15-04622-f002:**
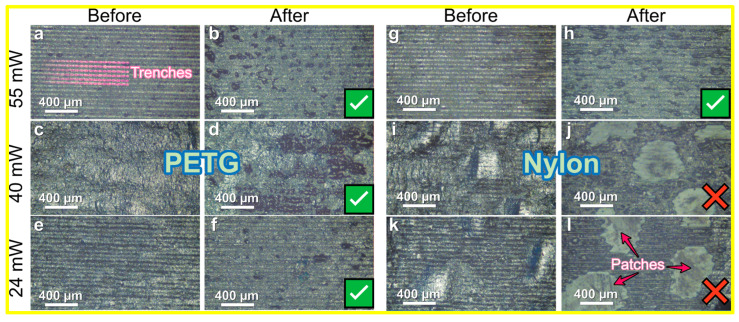
Optical microscopy images of the rGO films on PETG (**a**–**f**) and nylon (**g**–**i**) treated under different laser powers before (**a**,**c**,**e**,**g**,**i**,**k**) and after (**b**,**d**,**f**,**h**,**j**,**l**) washing in an ultrasound bath. The marks at the bottom right corner indicate the samples’ conductivity (check mark—conductive, cross mark—not conductive). Images for other polymers are presented in the SI.

**Figure 3 polymers-15-04622-f003:**
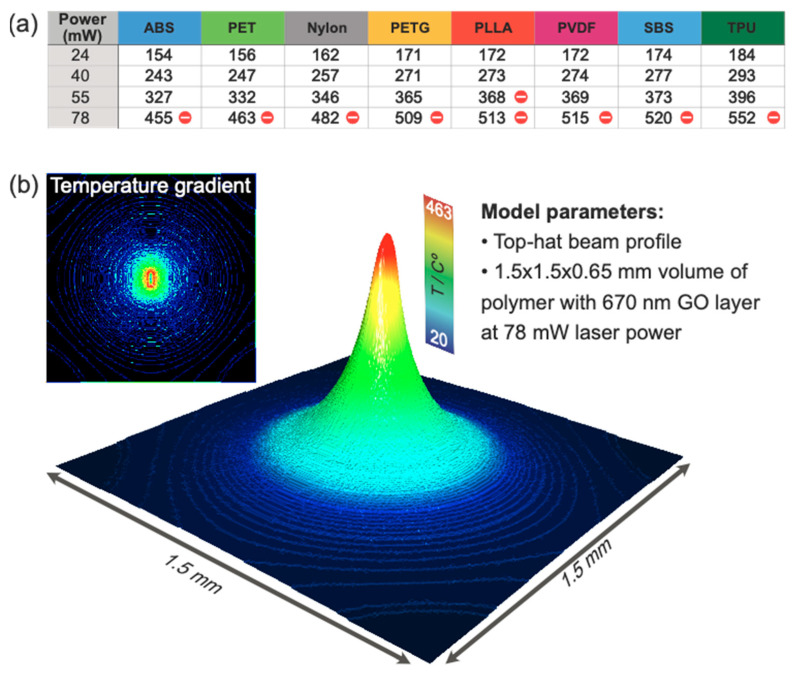
(**a**) Maximum polymer temperature in °C after 0.25 s of laser heating with different powers. The cells marked with red ticks indicate the destructive regime with a maximum temperature exceeding *T_d_*. (**b**) 3D image is a temperature map obtained by computer modeling of polymer heating under the laser treatment. The PET with GO film was taken as an example. In the inset is the temperature gradient map illustrating the high spatial confinement of temperature gradients responsible for mass flows due to the Marangoni effect.

**Figure 4 polymers-15-04622-f004:**
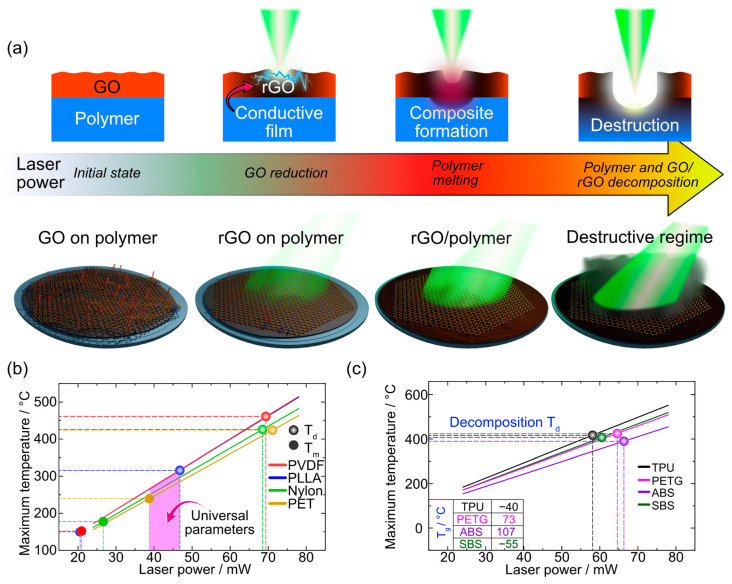
(**a**) Evolution of thermal processing on GO/polymer interfaces depending on laser power, (**b**) Simulation results of maximum temperature in the laser spot focused on GO-crystalline polymer interface vs. applied laser power. (**c**) Simulation results of maximum temperature in the laser spot focused on GO-amorphous polymer interface vs. applied laser power.

**Figure 5 polymers-15-04622-f005:**
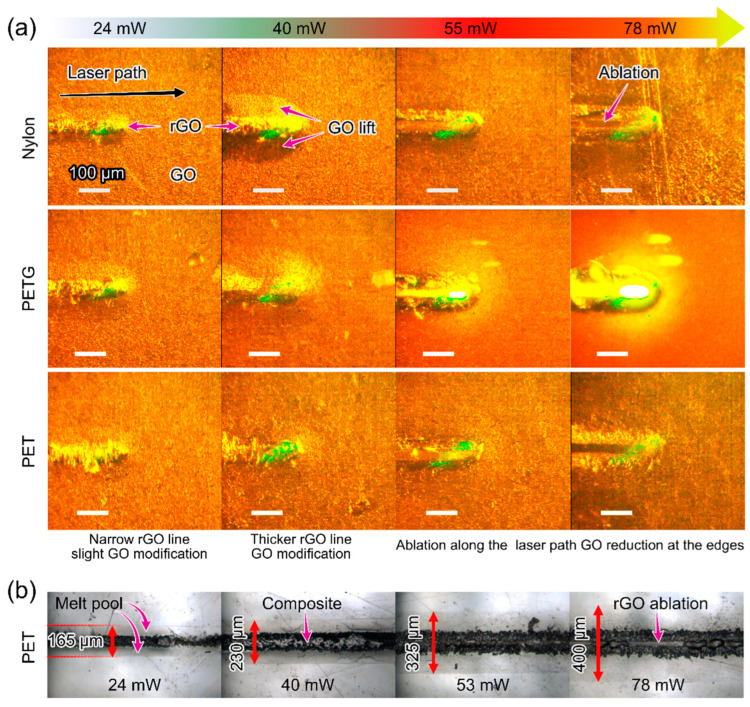
(**a**) High-speed shots recorded during laser processing GO films on polymers at different laser powers. The green flashes originate from the laser spot. The scale bar is 100 µm. (**b**) Optical microscopy images of the laser path on GO/PET after ultrasound cleaning show different dominant effects as laser power increases. The red arrows show the melt pool size that extends far from the laser-irradiated region. Images for all other polymers are presented in the SI.

**Figure 6 polymers-15-04622-f006:**
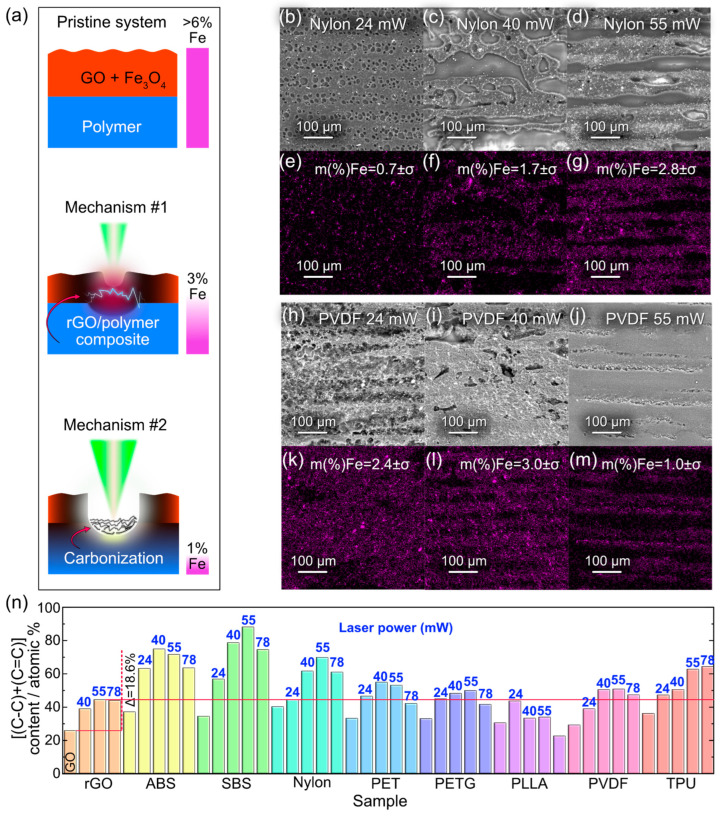
(**a**) Schematic representation of the initial system and post-processing mechanisms showing the pristine GO + Fe_3_O_4_ on polymer, followed by the possible rGO/polymer composite formation (Mechanism #1) and carbonization (Mechanism #2) with corresponding iron content. (**b**–**d**) Nylon-based samples microstructure evidenced by SEM imaging. (**e**–**g**) Elemental analysis mapping iron content by EDX of nylon-based samples. (**h**–**j**) PVDF-based samples microstructure evidenced by SEM mapping. (**k**–**m**) Elemental analysis mapping iron content by EDX of PVDF-based samples (measurement error *σ* < 0.1%). (**n**) The sum of atomic content of C–C and C=C bonds in the C-region of rGO/polymer samples. The laser power (in mW) used for each sample is shown in blue color. The red line shows the maximum value obtained for rGO/glass.

**Figure 7 polymers-15-04622-f007:**
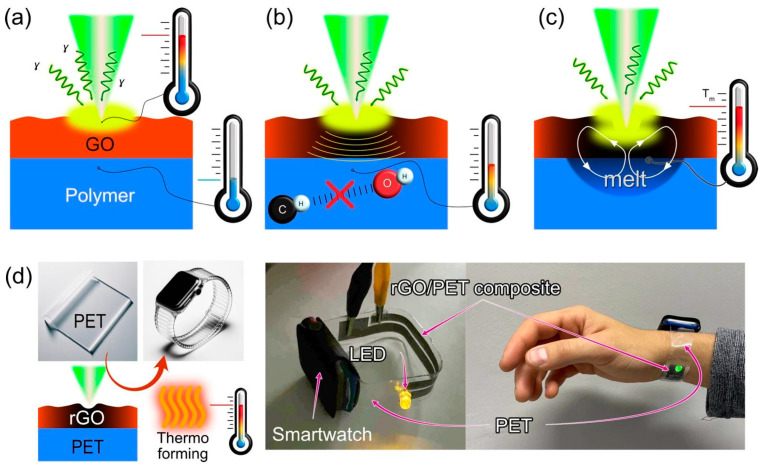
The processes occurring during GO on polymer laser processing. (**a**) Heating of GO and its reduction. (**b**) Polymer heating and destruction of H-O-H bonds. (**c**) Polymer melting and the occurrence of convective flows. (**d**) Demonstration of thermoformed wearable wristband, proving the possibility of using rGO/polymer composite in flexible electronics.

## Data Availability

The data presented in this study are available on request from the corresponding author.

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
