# Peer review of "Universal Approach to Integrating Reduced Graphene Oxide into Polymer Electronics"

_polymers, 2023, doi:10.3390/polym15244622_

Round 1
Reviewer 1 Report
Comments and Suggestions for Authors
In this work, the authors integrated reduced graphene oxide into polymer electronics, determined a specific laser power density (range between 0.8 and 1.83 kW/cm2) can obtain a large number of thermoplastic polymer graphene polymer composite coating, and by rGO laser induced integration and heat formation prepared a conductive wristband for wearable smart watches. However, the following issues need to be addressed before considering publication.
1. The keywords (e.g. flexible electronics, wearable electronics) is not much related to the abstract, so please reconsider the selection of keywords.
2. The authors are requested to detail the innovation of this work and whether it is somewhat universal.
3. The author's description of the work concept diagram (Figure 1) is relatively general, making readers unable to understand the specific mechanism.
4. The discussion part should be studied in depth, besides illustrate the experimental phenomena.
5. For the demonstration of the thermoformed wearable wristband, the author can provide a specific video for the reader to understand more intuitively.
Comments on the Quality of English LanguageMinor editing of English language required
Reviewer 2 Report
Comments and Suggestions for Authors
The manuscript entitled " Universal Approach to Integrating Reduced Graphene Oxide into Polymer Electronics" identified a particular range of laser power densities for generating graphene polymer composite coatings. The experimental parameters are explained using numerical simulations using conductivity as the index. SEM and High-speed videos are used for characterization. This study shed light on wearable electronics manufacturing thus promoting the application and potentially open the way for manufacturing 3D architected conductivity polymer structures.
I recommend publication in Polymers under minor revision.
1. is there any condensation or agglomeration of rGO under the laser melting process?
2. Can the authors elucidate more on why Nylon doesn't follow the proposed temperature law of forming a conductive composite?
3. Since dropcasting is used to deposit GO onto the polymer substrate, nature agglomeration will occur during the drying, a phenomenon is called "Coffee Ring" effect, this agglomeration can greatly reduce the dispersion uniformity on the polymer surface. How do the authors overcome this issue?
Moreover, correspondingly, can the authors elucidate more on why using 700nm~ GO for the simulation, how do the authors mesh this thin layer GO? It looks like there is no cross-section (thickness direction) thermal transport study
Comments on the Quality of English LanguageEnglish is fine, and no obvious spelling error is detected.
